# Optimizing Memory Placement using Evolutionary Graph Reinforcement Learning

**Shauharda Khadka** [*]
Intel Labs

**Estelle Aflalo** [*]
Intel Israel

**Mattias Marder** [*]
Intel Israel

**Avrech Ben-David** [*]
Technion

**Santiago Miret**
Intel Labs

**Shie Mannor**
Technion

**Tamir Hazan**
Technion

**Hanlin Tang**
Intel Labs

**Somdeb Majumdar** [†]
Intel Labs

## Abstract

For deep neural network accelerators, memory movement is both energetically expensive and can bound computation. Therefore, optimal mapping of tensors to memory hierarchies is critical to performance. The growing complexity of neural networks calls for automated memory mapping instead of manual heuristic approaches; yet the search space of neural network computational graphs have previously been prohibitively large. We introduce Evolutionary Graph Reinforcement Learning (EGRL), a method designed for large search spaces, that combines graph neural networks, reinforcement learning, and evolutionary search. A set of fast, stateless policies guide the evolutionary search to improve its sample-efficiency. We train and validate our approach directly on the Intel NNP-I chip for inference. EGRL outperforms policy-gradient, evolutionary search and dynamic programming baselines on BERT, ResNet-101 and ResNet-50. We additionally achieve 28-78% speed-up compared to the native NNP-I compiler on all three workloads.

## 1 Introduction

The proliferation of deep learning (DL) has been fueled, in part, by a rapid growth in the size and complexity of deep neural networks (DNN) (Dean et al., 2012; Ying et al., 2018). This has spurred the rapid development of hardware (Wang et al., 2016; Jouppi et al., 2017) and software (Abadi et al., 2016; Paszke et al., 2018; Cyphers et al., 2018) dedicated to deep learning workloads that seek to optimize critical performance metrics like throughput and power efficiency (Mattson et al., 2020). Producing compiler optimizations that map the tensors of a neural network's computational graph to the memory units on host hardware is a critical challenge. Since different memory types trade off bandwidth and capacity differently, a sub-optimal mapping could significantly increase latency.

For DL inference, the computational graph is static and placement can be pre-planned instead of relying on online cache management (Zhang et al., 2020; Shi et al., 2019). However, this is especially challenging with DNNs due to the high dimensional search space. For example, ResNet-50 (He et al., 2016) has 57 operational layers. Mapping each activation and weight tensor to, for example, three (DRAM, LLC, and SRAM) memory caches represents $3^{(2*57)} \approx 10^{54}$ possible decisions. BERT (Devlin et al., 2018) has 376 operational layers, and a search space of $\sim 10^{358}$. Since optimizing this mapping is intractable with traditional approaches, such as dynamic programming (Bellman, 1954), current solutions primarily rely on manually-tuned heuristic rules encoded in a compiler.

Because of the large search space, prior reinforcement learning (RL) algorithms for automating mappings have relied on manually-designed grouping (Mirhoseini et al., 2017; Addanki et al., 2018) or a learned grouper whose hierarchical structure is domain dependent (Mirhoseini et al., 2018). In addition to the extremely large action space, the large number of nodes render the reward sparse and noisy, and thus further unsuitable for gradient-based Deep RL algorithms. This sparsity stems from the fact that an overall performance metric can only be measured after all nodes have been processed.

---

[*]Equal Contribution
[†]Correspondence to: <somdeb.majumdar@intel.com>

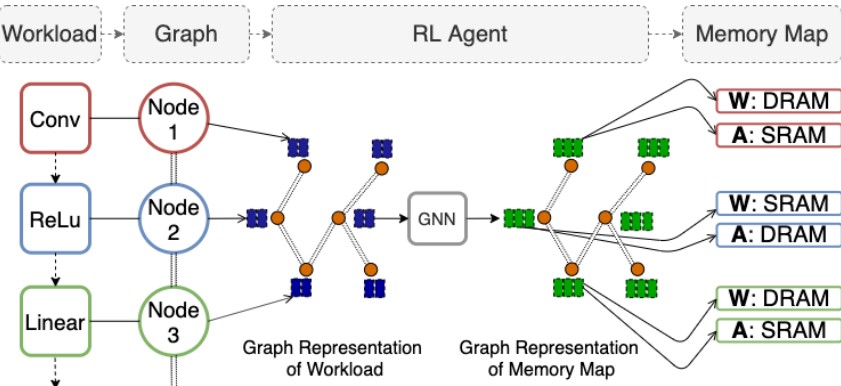

Figure 1: Workflow of Graph RL agent mapping weights (W) and activations (A) of each layer of a trained neural network workload to various on-board memory components (e.g. DRAM, SRAM).

In this paper, we present Evolutionary Graph Reinforcement Learning (EGRL), a hybrid approach of evolutionary search with gradient-based learning, that is able to natively search in a high-dimensional space that is orders-of-magnitude larger than previous approaches. EGRL is an extension of CERL (Khadka et al., 2019), a population based method for sparse-reward tasks that combines fast policy gradient (PG) learning with a stable evolutionary algorithm (EA). Since the action spaces explored in this paper are several orders of magnitude larger than those explored in CERL, we introduce Boltzmann chromosomes - a set of fast, stateless policies that accelerate evolution by providing partially optimized solutions as anchors. This mechanism is necessary to improve the sample-efficiency of the slow EA component for this large action space. Further, we employ a graph neural network (GNN) (Wu et al., 2020; Scarselli et al., 2008) to represent our policy. This allows our agent to natively process computational graphs representing deep learning workloads, enabling generalization over workloads of varying size and connectivity.

We demonstrate our solution on the Intel Neural Network Processor for Inference (NNP-I), a deep learning accelerator, to map modern neural networks on one of the three memory hierarchies on the chip. Each memory level in this chip has trade-offs in memory size and bandwidth, as detailed in Wechsler et al. (2019). This additionally differentiates our work from prior works such as REGAL (Paliwal et al., 2020) that assume infinite bandwidth and memory that are not practical on real hardware. Additionally, we consider single-batch inference, an important industry benchmark Mattson et al. (2020). While large batch sizes have greater computational efficiency (e.g., Boudoukh et al. (2020) on NNP-I), they are sub-optimal for a given inference example due to the latency associated with queuing up a batch. Therefore, single-batch inference is key to many time-critical applications (Park et al., 2018) where an individual inference query needs to be processed in real-time.

Results on ResNet-50, ResNet-101 (He et al., 2016) and BERT, show that EGRL significantly outperforms the chipset's native compiler across all workloads, and exceeds the performance of dynamic programming, evolutionary search and policy-gradient approaches.

Specifically, the contributions of this work are:

1. A generalized GNN-based policy that can natively accept a computational graph and produce a corresponding graph representation with the optimal memory maps. This eliminates the need for serialized, layer-dependent representations.

2. EGRL, a scalable population-based algorithm that can effectively train on sparse and noisy feedback from the host hardware in large search spaces.

3. An RL agent that trains directly on the hardware, with a feedback mechanism for constraint violation, and thus allowing direct deployment and testing on hardware.

## 2   RELATED WORK

**Optimizing Hardware using Machine Learning:** In this work, we study the problem of mapping tensors to memory components on a device. Several recent works have studied the use of machine learning to optimize the execution of computation graphs on hardware -a similar combinatorics problem. Mirhoseini et al. (2017) designed a policy gradient (PG) based *Placer* policy to map parts of neural models on hardware. However, it relied on a heuristic grouping strategy to significantly reduce the action space. Mirhoseini et al. (2018) improved this architecture by replacing the heuristic module with a *Grouper* policy. While this does represent an end-to-end PG approach, it is significantly more complex and hyperparameter-heavy than EGRL's PG network. Specifically, their *Placer* is an LSTM based Seq-to-Seq model with attention. The hierarchical structure is domain dependent - specific to operation grouping. Mirhoseini et al. (2020) also applied Deep RL to learn subsystem placement to optimize power, performance and chip area but relied on similar heuristic grouping strategies to significantly reduce the action space seen by the agent.

A closely related work is Placeto (Addanki et al., 2018) where the nodes of a computation graph are sequentially placed on hardware. Similar to EGRL, they also operate on a GNN representation. EGRL primarily differs from Placeto in simultaneously mapping all nodes. We found empirically that a sequential mapping strategy was significantly more sample-inefficient and could not scale to larger workloads. Sequential mapping strategies have the additional disadvantage of not being able to exploit parallelism during policy training. Placeto also adopts the manual grouping strategy from Mirhoseini et al. (2018) and adds an additional *Merge and Collocate* heuristic.

In contrast, EGRL has a simpler architecture using generic PG and EA to scale to large search spaces. For example, previous work with manual grouping operate at most in $5^{280} \approx 10^{196}$ dimensional action space (Mirhoseini et al., 2017), compared to $\sim 10^{358}$ for our BERT problem. Compared to pure PG based approaches, EGRL has significantly fewer hyperparameters to tune - primarily due to the reduced dependency on PG learning by using a population based search to handle sparse rewards.

Another closely related work is REGAL (Paliwal et al., 2020), which optimizes run-time and peak-memory via hardware placement. It also utilizes a graph representation with a genetic algorithm (GA) guided by RL. The RL agent predicts the parameters of GA - a form of indirect information transfer - while GA directly optimizes the final strategy. In contrast, our RL and EA components co-optimize the mapping strategies via direct information transfer (policy migration) and a shared replay buffer. REGAL assumes infinite bandwidth, whereas we train and validate entirely on physical hardware introducing specific mechanisms to incentivize compiler-valid mappings. This ensures that our solutions are performant under real-world operating conditions and closer to production-use.

As a relatively new research field, we are challenged by the unavailability of reproducible code for prior work. The domain specific heuristics as described above render it difficult to apply algorithms designed for, say, chip placement to our memory mapping problem. Therefore, we adopt a state-of-the-art PG method as a baseline, since PG is a common central component of the above prior work.

**Classical Search Methods:** Classical methods such as Simulated Annealing (SA) (Kirkpatrick et al., 1983) and genetic algorithms (GA) have also been studied for problems that have a similar combinatorial search complexity as memory placement. SA evolves new solution via small perturbations on existing solutions and retaining solutions that yield improved performance. A temperature parameter drives the exploration into new solution spaces. Our evolutionary algorithms (EA) (Floreano et al., 2008; Lüders et al., 2017; Fogel, 2006; Spears et al., 1993) improve on SA by systematically evolving new solutions within a population of solutions by performing mutations (similar to SA) and cross-over between pairs of solutions. Both methods are known to produce highly performant and stable solutions but are also significantly slower compared to Deep RL.

In this work, we use EA both as a component of EGRL and also as a baseline. The PG components of EGRL produce fast, semi-performant solutions which then become anchors in the EA module. This essentially "guides" the EA to a performant solution by providing better anchors to search around. We demonstrate this via ablation studies that isolate the EA and PG components of EGRL. We also introduce Boltzmann chromosomes in the EA population - a set of stateless policies that directly perturb action proposals to accelerate exploration – with a temperature term that balances exploration and exploitation. This component is motivated by SA.

**Evolutionary RL:** Our overall architecture builds on top of CERL (Khadka and Tumer, 2018; Khadka et al., 2019) which combines EA and PG. It diversifies exploration by allowing a population of EA policies to add data to a central replay buffer shared by a population of PG learners. We directly build on CERL because it has been shown to be effective in optimizing sparse feedback signals. Our memory mapping solution inherently relies on optimizing a very sparse feedback signal (e.g., latency) that is obtained at the end of an inference run on a workload.

For the PG portion of our architecture, we adopt Soft-Actor Critic (SAC) (Haarnoja et al., 2018), the current state-of-the-art model-free algorithm developed for continuous high-dimensional settings. SAC uses an actor-critic architecture with separate networks for the policy and the Q-value function. A stochastic Gaussian policy enables it to use a maximum entropy objective (Ziebart et al., 2008) through which it demonstrates state-of-the-art results. We modify SAC to be compatible with our discrete action space.

## 3 METHOD

We formulate the hardware mapping problem as a Markov Decision Process (MDP) and apply RL to train a solution. Figure 1 illustrates the high-level formulation of our workflow. A trained neural network (e.g., ResNet-50) is referred to as the *workload*. We convert the network to a graph representation which acts as the input to a GNN-based RL policy. The RL agent takes this graph input and proposes a memory location for the weight and activation tensors corresponding to each layer in the input workload, with the policy goal being to maximize a performance metric. The RL agent is implemented as a Graph U-Net policy based on Gao and Ji (2019). Algorithm 1 details the interaction of the agent with the environment.

In order to evaluate any policy, we run an inference call on hardware using the memory map proposals by replacing the compiler's native memory mapping module with our policy.

**State:** We convert the workload to a directed graph representation where each node represents an operational layer (conv, pooling etc) and the edges denote the connectivity between them. A detailed description of the node features can be found in Appendix A. Since all the outgoing edges of a node denote the same output tensor, their associated information are encapsulated in their source node features, leaving the edges featureless. The graph represents all mappable tensors (nodes) simultaneously - and thus allows us to have a parallelized training pipeline. This compute advantage is a primary reason for this design choice compared to serialized representations adopted in some prior works.

**Actions:** Each node contains information about two tensors, for the weights and the activations, that each must be individually mapped to one of three memory units (DRAM, LLC, SRAM) on the chip. The RL agent accepts the input graph and produces an output graph with identical topology, where the node features of the output graph represent the two memory location proposals to map the weight and activation tensors corresponding to the node. The memory hierarchy of NNP-I is detailed in (Wechsler et al., 2019), with a high capacity (32GB) DRAM, a lower capacity (24MB) but faster ($\sim 10\times$) LLC, and a small SRAM (4MB) per chip that is very fast ($\sim 100\times$) compared to DRAM.

The agent's complete memory map $\mathcal{M}_\pi$ is then sent to the compiler. If any of the mapping decisions cannot be executed on the hardware (i.e., invalid mapping), the compiler rectifies them and outputs a modified map, $\mathcal{M}_\mathcal{C}$, that is fully executable (Line 6 in Algorithm 1).

**Rewards:** In a standard RL setting, one can generally constrain the action space to avoid invalid actions. However, in our problem setting, constraining the action explicitly requires reproducing the compiler's logic for valid mappings, which would vary across hardware and compiler versions. In order to keep the RL algorithm independent of the compiler logic, we formulate separate reward domains for invalid and valid mappings. If the agent produces any invalid mapping that the compiler re-assigns, we do not execute an inference. Instead, we formulate a negative reward as a function of the re-assigned bytes ratio, to quantify the extent of the invalidity (Line 12 in Algorithm 1). This formulation allows us to avoid implementing the compiler logic explicitly - instead relying on a negative feedback that enables the agent to implicitly learn the rules for valid mappings. When the agent does produce a fully valid mapping, we execute inference and compute a positive reward. This reward is a function of the agent performance score normalized by that of the native compiler (Line 10 in Algorithm 1). While the normalization is not necessary when training on a single workload,

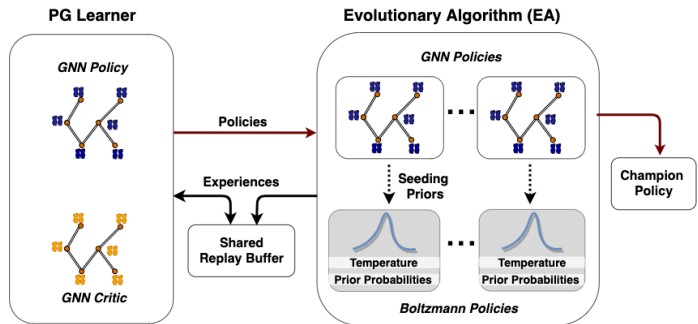

Figure 2: EGRL Architecture: EA and PG operate concurrently via a shared replay buffer. EA comprises sub-populations of GNN and Boltzmann policies. PG policy periodically migrates to EA.

it allows for flexibility to concurrently train on multiple workloads that might have a wide range of scores. For our application, we maximize the reciprocal of latency as it is an intuitive measure of speed and easily implemented.

**Training:** Our training algorithm, EGRL, builds on the CERL framework (Khadka et al., 2019) to tackle variable-sized, multi-discrete action settings. Figure 2 illustrates the high level architecture of EGRL. It is comprised of a single PG learner, with a GNN architecture, and an EA population containing a mixture of GNN and stateless Boltzmann policies. Each individual in the population provides a different mapping proposal for a given input workload. These proposals are evaluated by performing an inference run on the input workload using the proposed mappings and measuring the resultant latency. During the evaluation process, all data generated by all policies is stored in the PG learner's replay buffer.

| **Algorithm 1** Agent's Interaction with the Environment |
| --- |

1:  **Initialize** workload $f$, policy $\pi$, perf. metric $\Omega$
2:  **Initialize** compiler $\mathcal{C}$ and graph transform $\mathcal{G}$

3:  $\mathcal{G}(f) \leftarrow f$         ▷ Workload to graph
4:  **for** each iteration $i$ **do**
5:      $\mathcal{M}_\pi = \pi_i(\mathcal{G})$        ▷ Agent's map
6:      $\mathcal{M}_\mathcal{C} = \mathcal{C}(\mathcal{M}_\pi)$     ▷ Compile agent's map
7:      $\epsilon_\mathcal{M} = \mathcal{M}_\pi \| \mathcal{M}_\mathcal{C}$       ▷ Mapping error
8:      **if** $\epsilon_\mathcal{M} == 0$ **then**
9:         $\Omega = \mathcal{I}(\mathcal{M}_\mathcal{C})$       ▷ Run inference
10:        $r_\mathcal{M} = (\frac{\Omega}{\Omega_{baseline}})^2$    ▷ Positive reward
11:     **else**
12:        $r_\mathcal{M} = -\epsilon_\mathcal{M}$      ▷ Negative reward
13:     $\pi_{i+1} \leftarrow \pi_i$        ▷ Update policy

The population then undergoes standard EA processes to produce a new generation of candidate policies.

Concurrently, the PG learner updates its actor and critic by sampling from the shared replay buffer, as is typical for off-policy algorithms. The PG learner is periodically copied into the EA population as a form of information transfer - and is allowed to participate in evolution. At any given time, the top-ranked policy in the EA population is chosen for deployment. We adopt SAC as our PG algorithm and modify it for our discrete action space setting. A detailed description of our algorithm can be found in Appendix C, with our modifications to SAC described in Appendix D.

The Boltzmann chromosome is an additional policy representation we introduced into the population. Each Boltzmann chromosome is parameterized by a set of prior probabilities ($P$) and a temperature ($T$) for each node. To compute an action for each node, we sample from the Boltzmann softmax function (Asadi and Littman, 2017) using that node's $P$ and $T$, making it significantly faster to compute an action compared to a neural network policy. A higher temperature $T$ results in higher exploration by selecting decisions farther from $P$. Crucially, $T$ is learned (via evolution) for each node independently which allows for varying degrees of exploration-exploitation across different mapping decisions simultaneously. A longer discussion on the Boltzmann chromosome is included in Appendix E.

The EA population concurrently holds both GNN and Boltzmann policies. All policies share data and benefit from the joint exploration. The PG based GNN policy can directly leverage the states explored by the Boltzmann policy to compute gradients. Conversely, as shown in Figure 2, the Boltzmann

policy's prior $P$ is periodically seeded using the GNN policy's posterior probability distribution - thus enabling it to directly bootstrap from the GNN population.

# 4 EXPERIMENTS

We evaluated the performance of EGRL and baseline implementations on Intel NNP-I hardware. For a given DNN workload, our agents controlled how their intermediate tensors are mapped to memory units on the chip. We then report the resulting latency as measured directly in the hardware. We conduct both training and testing entirely on the physical hardware. A complete set of hyperparameter details can be found in Appendix B and our code will be open-sourced.

**Workloads Tested:** We benchmarked our algorithms on three popular neural network workloads. ResNet-50, with 57 nodes, is widely used for benchmarks such as MLPerf (Reddi et al., 2019). ResNet-101, with 108 nodes, allowed us to test our algorithms at greater scale. Lastly, BERT, with 376 nodes, is a state-of-the-art natural language processing model. This allowed us to test for scale and generalization of our approach. Since the action space for a workload with $N$ nodes is $3^{(2N)}$, the corresponding sizes of the action spaces are $3^{114} \approx 10^{54}$ (ResNet50), $3^{216} \approx 10^{103}$ (ResNet101) and $3^{752} \approx 10^{358}$ (BERT) respectively.

**Metrics Reported:** We define *speedup* as the relative improvement in latency achieved by the agent's mapping versus that of the compiler. A score greater than 1 indicates an improvement in latency while a score between 0 and 1 indicates a degradation. A score of 0 indicates an invalid mapping. We conduct 5 independent statistical runs and report the mean and standard deviation. Further, we report all speedups against iterations where an iteration refers to an inference process on the physical hardware. We report the iterations cumulatively across the population. Since policies are updated after each iteration, the number of iterations is equivalent to sample-complexity.

**Baselines:** We use the Intel NNP-I's default compiler mapping as our primary baseline. The compiler consists of a collection of heuristic rules specific to the compute and memory capacity of the hardware. This is common to many other hardware subsystems. In all our runs, we simply replace the compiler's memory mapping module with our policy while keeping all other processes, including low-level cache management, the same.

We evaluate a pure PG approach by testing the modified SAC-discrete algorithm. This is motivated by prior works like Mirhoseini et al. (2018) and Mirhoseini et al. (2017) which have shown effective results on similar hardware mapping problems using PG. We also implement an EA baseline given its importance as a classical search method. Since PG and EA are also core components of EGRL, these baselines also serve as ablation studies to distil the relative contribution of each component to the final performance improvements.

Finally, we implement a **Greedy Dynamic Programming (DP)** agent, inspired by classical DP methods for optimization (Andonov et al., 2000; Bertsimas and Thiele, 2004). It makes layer-wise greedy decisions directly on the workload. It tries all 9 possible maps for the first node (3 choices each for 2 types of tensors), keeping all other mappings static, and chooses the action that maximizes the reward. It repeats this serially for each node in the workload and then conducts several passes. It essentially assumes conditional independence of mapping across $N$ nodes to reduce the solution space from $9^N \rightarrow 9 * N$. While this is a fairly naïve assumption, running multiple passes through the graph produces a reasonable solution.

# 5 RESULTS

We show the speedup achieved, relative to the compiler and measured directly on the NNP-I chip, for the various agents tested on the ResNet-50, ResNet-101 and BERT in Figure 3. EA and EGRL significantly outperform the compiler across all three workloads. Greedy-DP approaches baseline performance while the PG agent in isolation fails to reach it at all.

On ResNet-50, EGRL and EA significantly outperform the baseline compiler and all other agents reaching a final speedup of 1.28 and 1.06, respectively. Greedy-DP underperforms the compiler at 0.72 while PG converges to 0.29. On ResNet-101, EGRL again significantly outperforms the baseline compiler and all other agents reaching a final speedup of 1.78 while EA converges to 1.47.

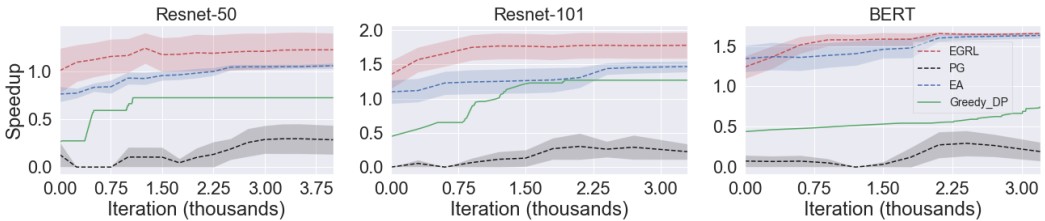

Figure 3: Speedup for different workloads, normalized to the heuristic compiler performance. EGRL consistently outperforms all baselines. Error bars indicate standard deviation over $n = 5$ runs.

This performance gap demonstrates the role played by the collaborative learning using the shared replay buffer in EGRL. While the PG learner fails to find full mapping solutions by itself, the partial solutions it finds carry vital information to the EA population. Greedy-DP outperforms the compiler, converging to $1.27$, while PG alone converges to $0.23$.

On BERT, EGRL and EA significantly outperform the compiler and all other agents reaching a final speedup of $1.66$ and $1.64$, respectively. Greedy-DP converges to a speedup of $0.67$, greatly underperforming the compiler. This is unsurprising as BERT is comparatively much larger in size than the two ResNet models which breaks the Greedy-DP's conditional independence assumption. PG fails to find good mappings and converges to $0.21$.

**Generalizability Properties**: While EGRL performs well when trained on each workload individually, we investigated if the policy embeddings derived from learning on one workload were performant on a second workload. We tested this by taking a policy trained on one workload and evaluating it on a second workload with no further training. This simply involves taking the GNN policy learned on the first workload and replacing the input and output graph representations to be compatible with the topology of the second workload - keeping the hidden layers - and thus the learned embeddings - untouched. We tested this at various stages of training on a given first workload.

Figure 4 shows how policies trained on BERT and ResNet-50 transferred to the held-out workloads at different points in their training. While the transferred policies clearly underperform those trained from scratch, it is notable that they still outperformed the baseline compiler. While a full study of generalization and transfer learning is out of the scope of this work, this indicates that the GNN embeddings can be domain invariant. We also observe that the zero-shot performance drops after a certain point in training, which indicates the point at which the policy overfits to its training workload.

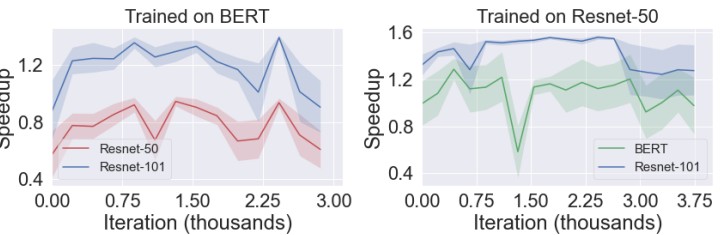

Figure 4: Zero-shot Generalization of policy embeddings: policies trained on one workload are tested on the others without fine-tuning

**Visualizing Memory Mappings:** We also studied the differences between the mapping strategies of the compiler and EGRL at different stages of training. For ease of analysis, we convert the mappings to a two-dimensional UMAP embedding (McInnes et al., 2018) as shown in Fig 5. For each workload, we collected its mappings twice - first when the agent's mappings approximately reach the compiler's speedup performance ($\sim 1$) denoted as *compiler-competitive-mappings*, and second when the agent reaches its best recorded speedup denoted as *best-mappings*.

The *compiler-competitive-mappings* and *best-mappings* are generally well-separable across all three workloads. Further, the compiler's mapping also fell within the cluster of compiler-competitive-mappings across all three workloads. This suggests that the agents learn to mimic the compiler's mappings at some point in their training. This is unsurprising as the reward we use to train the agents before they find valid mappings is based on differences with the compiler. Interestingly, the

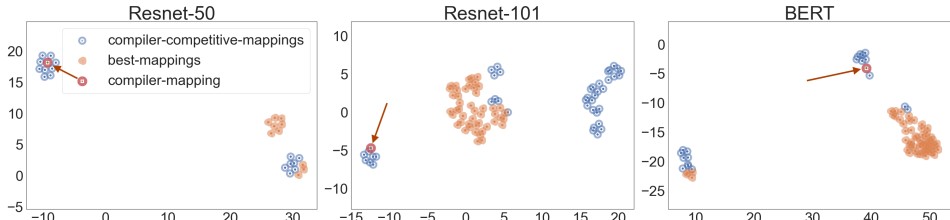

Figure 5: UMAP projection illustrating mappings that achieve compiler-competitive performance (speedup of $\sim 1$), the best mappings, and the compiler's mapping (highlighted with a red arrow).

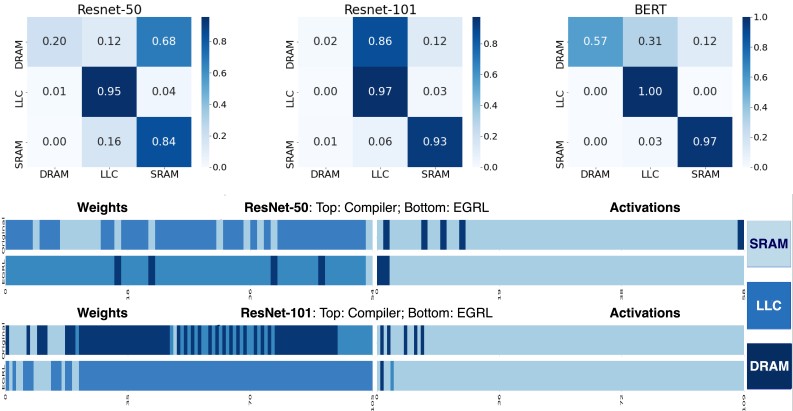

Figure 6: Memory map shifts **Top**: For each memory unit on the y-axis, the corresponding row shows how EGRL changed the distribution of tensors originally mapped to it by the compiler. **Bottom**: Memory maps from the compiler vs the best ones found by EGRL for ResNet-50 and ResNet-101. Each bar denotes a tensor operation.

intra-cluster spread for *compiler-competitive-mappings* is markedly higher than *best-mappings* across all three workloads. This indicates that the mappings associated with higher speedups are more self-similar than those that are less performant, which is also unsurprising since the number of inferior mappings is higher than that of the superior ones.

Figure 6 illustrates the differences in raw mappings between the compiler and EGRL. The transition matrices on top show how the distribution of tensors to the different memories shifted. Each row corresponds to a memory unit. The corresponding columns indicate how EGRL fractionally redistributed tensors originally mapped to that unit into all available memories. At the bottom, we illustrate how each tensor in a workload was mapped by the compiler and by EGRL. Each band represents either a weight or an activation tensor.

While it is difficult to semantically interpret the mapping decisions reliably, we observe that EGRL generally found memory maps that avoided the slower, but higher-capacity DRAM. This difference is particularly prominent for the weight tensors. EGRL also favored contiguity - where tensors from neighboring layers generally got mapped to the same type of memory. Both are performant strategies to optimize latency - but not trivial to achieve using heuristics that need to trade-off speed and capacity for a large number of tensors. One hypothesis is that EGRL's graph-based global view of the workloads enables it to make globally optimal allocations compared to the sequential decision making of the compiler.

## 6 DISCUSSION AND FUTURE WORK

This paper introduced EGRL, a hybrid framework to learn effective memory mapping solutions for large deep learning workloads. We train our policies end-to-end on the NNP-I chipset to ensure that

the solutions are robust to the real-world constraints and uncertainties of the chip. We show that EGRL scales effectively across varying sizes and operational types of DL workloads, and exhibits some zero-shot transferability across workloads. Results show that EGRL outperforms several learning and classical search and optimization methods as well as the heuristic logic of the compiler. By combining evolutionary and gradient-based approaches, and including stateless policies to accelerate evolution, we efficiently tackle the large search space of this problem. This scalability paves the way for learning-based agents to tackle other hardware mapping problems. Specifically, future work will expand the action space of the EGRL agent to control multivariate settings like batch size, ring frequencies, power efficiency and data decomposition.

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

## APPENDIX

## A   GRAPH EMBEDDING

Table 1 details the features we used for the node embedding. These features encapsulate encapsulate information about the input and output tensors of the given operation, as well as summary information about future layers.

| Node Features | Description |
|---|---|
| $op_{id}$ | Operation id |
| $weight\_size$ | Size in bytes of the weights if exist, 0 otherwise |
| $ifm_x$ | Input feature map size on the x axis |
| $ifm_y$ | Input feature map size on the y axis |
| $ifm_z$ | Input feature map size on the z axis |
| $ofm_x$ | Output feature map size on the x axis |
| $ofm_y$ | Output feature map size on the y axis |
| $ofm_z$ | Output feature map size on the z axis |
| $ifm\_size$ | Total size of the input feature map ($ifm_x * ifm_y * ifm_z$) |
| $ofm\_size$ | Total size of the onput feature map ($ofm_x * ofm_y * ofm_z$) |
| $n\_ops\_left$ | Total number of operations after current $node$ |
| $n\_w\_left$ | Total number of weights from current $node$ to the last node |
| $groups$ | Number of groups - Convolution related parameter, set to 0 otherwise |
| $kernel_x$ | Kernel size on x axis - Convolution related parameter, set to 0 otherwise |
| $kernel_y$ | Kernel size on y axis - Convolution related parameter, set to 0 otherwise |
| $stride$ | Stride size - Convolution related parameter, set to 0 otherwise |
| $pad$ | Padding size - Convolution related parameter, set to 0 otherwise |
| $dilation$ | Dilation - Convolution related parameter, set to 0 otherwise |
| $batch$ | Input batch size |

Table 1: GNN Node Features

## B   HYPERPARAMETERS

Table 2 details the hyperparameters used in the paper.

| Hyperparameter | Range explored | Value used |
|---|---|---|
| GNN hidden layer size | [32, 64, 128] | 128 |
| GNN output layer size | [32, 64, 128] | 128 |
| GNN depth | 4 | 4 |
| Number of GNN attention heads | [1, 4] | 4 |
| # Steps per Episode | [1, 5, 10] | 1 |
| Initial mapping action | ['DRAM'] | 'DRAM' |
| Reward for invalid mapping | [-10, -1] | -1 |
| Discount Rate | [0.9, 0.97, 0.99] | 0.99 |
| EA population size | [10, 20] | 20 |
| PG Rollout size | [0, 1, 10] | 1 |
| Fraction of EA population that are Boltzmann | [0.1, 0.2, 0.5] | 0.2 |
| Total steps in the environment | [4000, 10000] | 4000 |
| Replay buffer size | [100000] | 100000 |
| Critic learning rate | [1e-3, 1e-4] | 1e-3 |
| Actor learning rate | [1e-3, 1e-4] | 1e-3 |
| Alpha (Entropy Coefficient) | [0.05, 0.1, 0.2] | 0.05 |
| Tau (Double-Q Network synchronization rate) | [1e-3] | 1e-3 |
| Batch size for PG | 24 | 24 |
| Reward scaling multiplier | 5 | 5 |
| Gradients steps per environment step | 1 | 1 |

Table 2: Hyperparameters

## C EGRL

---

**Algorithm 2** EGRL Algorithm

---

1:  Initialize a mixed population of $k$ policies $pop_\pi$
2:  Initialize an empty cyclic replay buffer $\mathcal{R}$
3:  Define a random number generator $r() \in [0, 1)$
4:  **for** generation = 1, $\infty$ **do**
5:      **for** actor $\pi \in pop_\pi$ **do**
6:          fitness, Experiences = Rollout($\pi$)
7:          Add experiences to $\mathcal{R}$
8:      Rank the population based on fitness scores
9:      Select the first $e$ actors $\pi \in pop_\pi$ as elites
10:     Select $(k-e)$ actors $\pi$ from $pop_\pi$, to form Set $S$ using tournament selection with replacement
11:     **while** $|S| < (k - e)$ **do**
12:         Select $\pi_a \in e$ and $\pi_b \in S$
13:         **if** $\pi_a$ and $\pi_b$ are of the same encoding type **then**
14:             Use single-point crossover and append to $S$
15:         **else**
16:             Sample a random state and get action $a$ from the GNN policy
17:             Use $a$ to encode the prior of the Boltzmann chromosome
18:     **for** Actor $\pi \in$ Set $S$ **do**
19:         **if** $r() < mut_{prob}$ **then**
20:             Mutate($\theta^\pi$) by adding noise $\sim \mathcal{N}(0, \sigma)$
21:     ups = # of environment steps taken this generation
22:     **for** ii = 1, ups **do**
23:         Sample a random minibatch of T transitions $(s_i, a_i, r_i, s_{i+1})$ from $\mathcal{R}$
24:         Update the critic via a Bellman update

25:         $L_i = \frac{1}{T} \sum_i (y_i - \mathcal{Q}_i(s_i, a_i^\sim))^2$
26:         where $y_i = r_i + \gamma \min_{j=1,2} \mathcal{Q}'_j(s_{i+1}, a_{i+1}|) + H(\pi(.|s_{i+1}))$

27:         where $a_i^\sim = a_i + \epsilon, clip(\epsilon \sim \mathcal{N}(\mu, \sigma^2) - c, c)$

28:         Update $L_\pi$ using the sampled policy gradient with noisy actions
29:         Soft update target networks:
30:         $L_{\theta^{\pi'}} \Leftarrow \tau L_{\theta^\pi} + (1 - \tau) L_{\theta^{\pi'}}$ and
31:         $L_{\theta\mathcal{Q}'} \Leftarrow \tau L_{\theta\mathcal{Q}} + (1 - \tau) L_{\theta\mathcal{Q}'}$
32:     Copy $L_\pi$ into the population: for weakest $\pi \in pop_\pi : \theta^\pi \Leftarrow L_{\theta^\pi}$

---

EGRL incorporates EA's population-based search with powerful gradient-based methods from DRL to expedite learning. In this work, we instantiate the EA population to use both the GNN encodings as well as a Boltzmann chromosome encoding to direct its search. Concurrently, we use a modified SAC Haarnoja et al. (2018) algorithm as our gradient-based technique in training the GNN policies. Algorithm 2 details the EGRL algorithm.

A general flow of the EGRL algorithm proceeds as follow: a mixed population of GNN-policies and Boltzmann-based policies is initialized with random weights. In addition to the population, one additional actor network (referred to as $pg_{gnn}$ henceforth) is initialized alongside a critic network. The population is then evaluated in the environment by allowing it to control the memory mapping for the specified workload in the given hardware. A selection operator then selects a portion of the population for survival with probability commensurate on their relative performance. The population is then probabilistically *perturbed* through mutation and crossover operations to create the next generation. A select portion top performers are preserved as elites and are shielded from the mutation step.

**Shared Replay Buffer:** Unlike a traditional evolutionary population, each individual (whether GNN or Botlzmann-based) stores its experience defined by the tuple *(current state, action, next state, reward)* in a globally shared replay buffer. This is done for every interaction that takes place with the hardware to maximize data efficiency. The critic samples a random minibatch from this shared replay

buffer and uses it to update its parameters using gradient descent. The critic is then used to train the $PG_{GNN}$ using the sampled policy gradient.

The shared replay buffer is a key mechanism that enables the sharing of information across the varying learning methods. In contrast to a standard search method which would extract the performance score and disregard the underlying data immediately, EGRL retains every interaction in the global buffer and engages the $PG_{GNN}$ and critic to learn from them repeatedly using powerful gradient-based methods. This enables maximal information extraction from each individual experiences as interfacing with the hardware is an expensive operation.

**Mixed Exploration:** A noisy version of the $PG_{GNN}$ using Gaussian noise generator is used to generate additional experiences for the replay buffer. In contrast to the population of GNN-actors which explore by noise in their neural weights, the $PG_{GNN}$ actors explore through noise in its *action space*. Boltzmann chromosomes tread this line in between where they explore in the parameters space more directly connected to the action space. Overall, each exploration technique are complementary and collectively lead to an effective exploration of the solution space.

**Migration:** Periodically, the $PG_{GNN}$ network's weights are copied into the evolutionary population. This process enables the evolutionary framework to directly leverage the information learned through gradient descent. This process also serves to stabilize learning and make it more robust to deception. If the policy learned by the $PG_{GNN}$ is favorable, it will be selected to survive and extend its influence to the population over subsequent generations. However, in case it is bad, it will be selected against and discarded. This mechanism ensures that the flow of information from the $PG_{GNN}$ to the evolutionary population is constructive.

## D   POLICY GRADIENT MODIFICATIONS TO SAC

**Policy Gradient Algorithm:** We build on SAC (Haarnoja et al., 2018) to tackle our large multi-discrete actions space. Since our policy is discrete, we compute entropy directly as

$$H(\pi(.|s)) = \mathbb{E}_{s \sim D}\big[ - \sum \pi(.|s) \log \pi(.|s) \big]$$

We then average over all nodes to compute the overall entropy of the policy. Further, we use a noisy version of the one-hot encoded behavioral action to compute our Bellman update as

$$L_i = \tfrac{1}{T} \sum_i (y_i - \mathcal{Q}_i(s_i, \widetilde{a}_i))^2$$
$$\text{where } y_i = r_i + \gamma \min_{j=1,2} \mathcal{Q}'_j(s_{i+1}, a_{i+1}|) + H(\pi(.|s_{i+1}))$$

We use the minimum of two heads from the Q-Network based on (Fujimoto et al., 2018). The noisy action $\widetilde{a}$ is computed by adding Gaussian noise clipped between $-c$ and $c$

$$\widetilde{a}_i = a_i + clip\big(\epsilon \sim \mathcal{N}(\mu,\, \sigma^2), -c, c\big)$$

This noisy action smoothens the value estimate towards similar state-action value estimates by the policy. It serves to make the policy smooth and addresses overfitting to the one-hot encoded behavioral output. The actor is trained using the sampled policy gradient.

## E   BOLTZMANN CHROMOSOME

Figure E.1 illustrates the operation of the Boltzmann chromosome for a particular action choice in one node. Parameters for prior ($p1$, $p2$, $p3$) and temperature $t$ fully encode the chromosome's policy. To compute an action, we first compute the probabilities by applying the Boltzmann softmax operation with the associated prior and temperature. Action is the sampled from this probability distribution. The choice

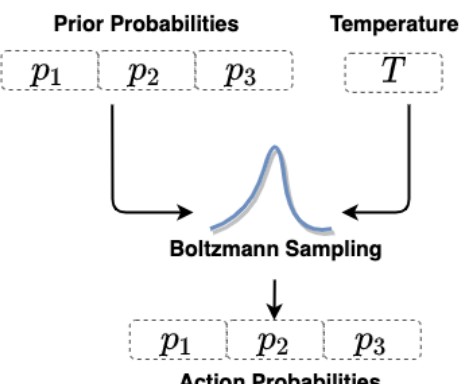

of temperature $t$ directly modulates the exploration-exploitation knob of decision making. A higher temperature leads to higher entropy probability distribution enabling higher exploration. In contrast, a lower value of temperature will lead to lower entropy in the probability distribution enabling exploitation of the prior information.

For the agent policy described in this paper, a Boltzmann chromosome solution comprises of priors and temperature parameters for each node and action choice in the computational graph. Learning either through seeding, mutation or crossover involves a direct update of these parameters. Importantly, these parameters are learned independently within the context of each node allowing for varying degrees of exploration-exploitation position across nodes. For instance, the agent could be very confident about mapping a specific node while concurrently be unsure for a different node of the same workload at the same time. The enables the agent to systematically balance the exploration-exploitation tradeoff at the resolution of individual node actions.

## F  MAPPING VISUALIZATIONS

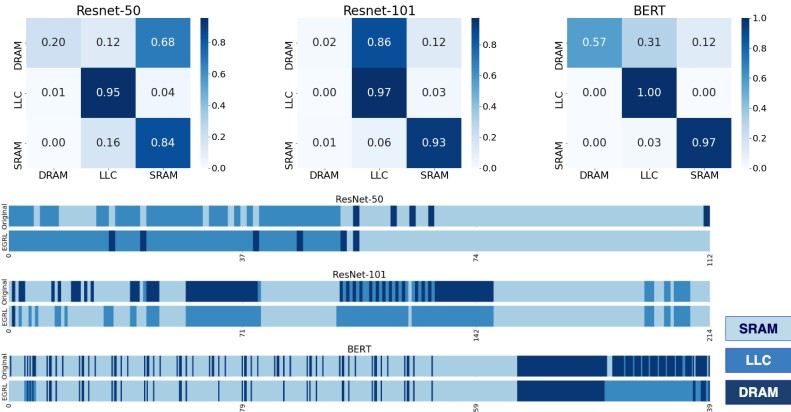

Figure F.2: Memory map shifts **Top**: For each memory unit on the y-axis, the corresponding row shows how EGRL changed the distribution of tensors originally mapped to it by the compiler. **Bottom**: Memory maps from the compiler vs the best ones found by EGRL for ResNet-50, ResNet-101 and BERT. Each bar denotes a tensor operation.

## G  NNP-I INFERENCE ACCELERATOR

NNP-I is hardware accelerator dedicated to neural network inference and was put in production in late 2019. It was designed to handle inference workloads at data center scale achieving 4.8 TOPs/W. The memory organization in NNP-I is shown in Figure G.3.

NNP-I includes twelve inference compute engines (ICEs) each having a 4MB Deep SRAM. A 24MB shared memory cache (LLC) is accessible by all ICEs. Additionally, a 32 GB DRAM is also accessible to the ICEs through the LLC. In terms of bandwidth, the DRAM is the slowest, while the LLC and Deep SRAM is 10x and 100x faster than the DRAM respectively.

In NNP-I, the specific mapping of each network layer to DRAM, LLC or SRAM is pre-determined statically during compilation. In our work, we intercept this capability of the compiler and design a reinforcement learning policy that performs this mapping. In this work, our RL agent only makes a gross determination if a tensor should be mapped to the DRAM, LLC or Deep SRAM memory components. If our agent specifies Deep SRAM as the recommended target for a tensor, the specific target out of the 12 possible Deep SRAM locations (one for each ICE) is left to the further logic in the compiler.

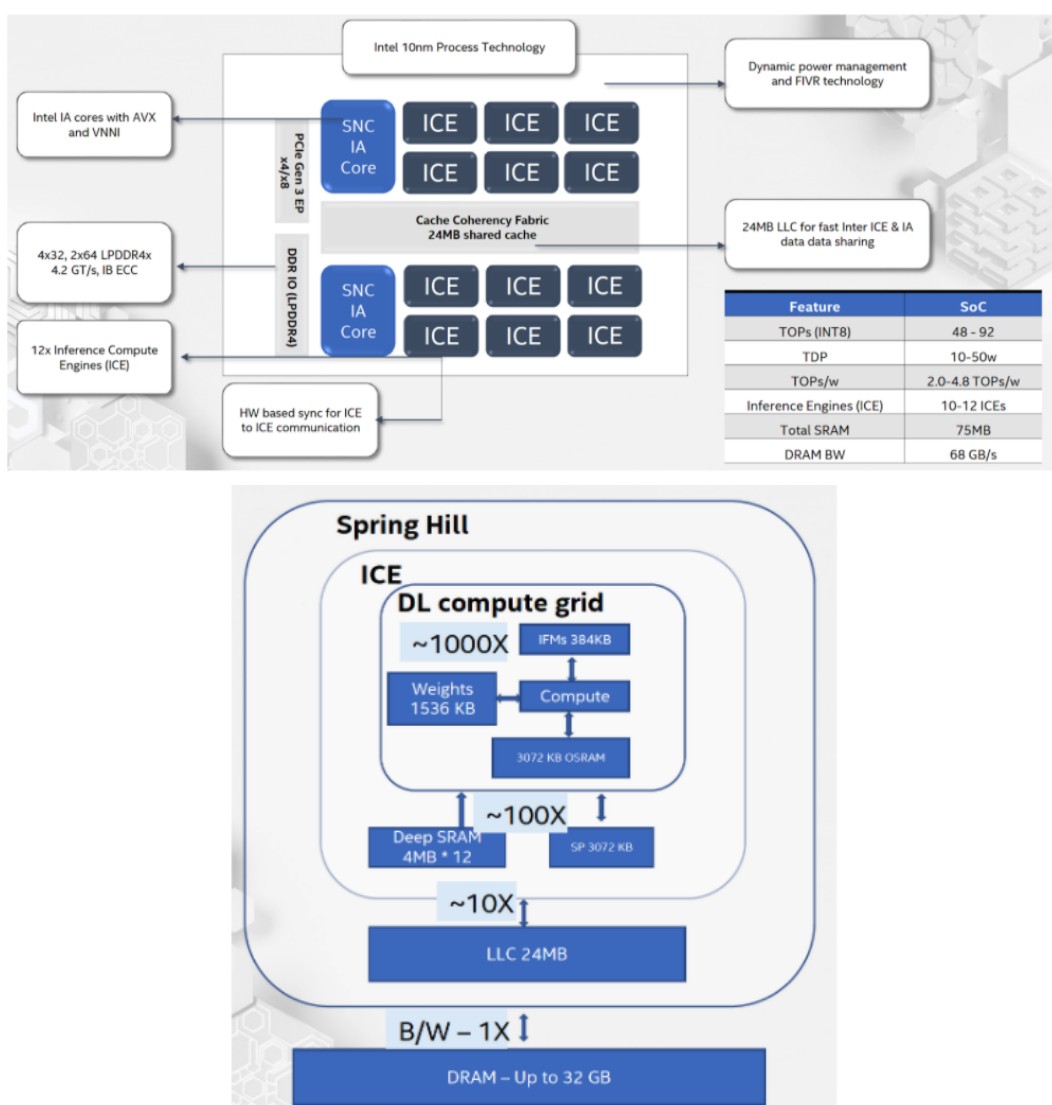

Figure G.3: Schematic overview of the NNP-I HW architecture. Characteristic for the NNP-I are its 12 Inference Computation Engines (ICE) each with its own 4MB Deep SRAM, the 24MB shared cache memory (LLC) accessible directly from all of the ICEs. Outside the LLC the NNP-I has a DRAM of 32GB accessible to the ICEs through the LLC.

## G.1 EGRL'S INTERACTION WITH NNP-I

Based on the architecture described above, EGRL can be utilized to dictate memory mapping at various control hierarchies. In this work, we designed EGRL to make mapping proposals for each tensor to the DRAM, LLC or SRAM classes of memory. It does not specify which of the multiple SRAMs a tensor should be mapped to. We let the downstream compiler optimization, including cache management strategies, control this data flow. An alternative would have been to expand the action space for EGRL to control for the number of active ICE cores, mapping tensors to specific ICE addresses, branch scheduling and other control logic typically residing in the compiler. While increasing this complexity has the potential for better overall performance, it also increases convergence time for the policy and, more pertinently, requires more domain knowledge of the underlying architecture. Thus, we chose the configuration that required minimal domain knowledge while discovering strategies that substantially outperformed the baseline.

Importantly, the GNN in EGRL only encodes the input workload explicitly. Since EGRL trains itself on the resultant latency, one could argue that it learns to model the consequence of the additional compiler optimizations implicitly - but this is never modeled explicitly. This is designed to require minimal understanding of the underlying architecture and data flows control.

Additionally, EGRL does not modify any aspect of the input workload. In that respect it is complementary to other works that optimize the graph itself - for example via quantization or model compression or graph re-writing methods as studied in (Ahn et al., 2020) where operational sequences that are sub-optimal for memory bandwidth are re-wired for optimal throughput. In all of these scenarios, EGRL is expected to achieve incremental performance improvements conditioned on the application-specific metric (e.g., throughput) and hardware-specific constraints.

