# OpenReview forum: "Optimizing Memory Placement using Evolutionary Graph Reinforcement Learning"
_ICLR.cc/2021/Conference — ICLR 2021 Poster_

### Official Review · AnonReviewer1 · 2020-10-26

**Rating:** 6
**Confidence:** 4

**Review:**

Summary:

This paper proposes a new algorithm called EGRL to improve computation graph running time by optimizing placement of the graph's components on memory. Specifically, the authors demonstrate the algorithm on the Intel Neural Networks Processor for Inference (NNP-I), which allows them to map neural network components on one of three memory hierarchies, each with different tradeoffs. The authors demonstrate that this technique provides speedups on BERT, Resnet-50 and Resnet-101.


Pros:

- Some past papers (for eg. [1]) in this domain evaluate their work in simulators instead of real hardware, and often, the simulators make assumptions that are not realisitc. The paper tests its technique on actual hardware, and this is definitely a plus.
- The authors promise that they will open-source their code. This is important since many of the efforts in this domain remain fragemented and difficult to reproduce. This is primarily due to the lack of open source code or a standard benchmark.
- The paper is well written.
- The visualisations of the learned policy vs the baseline in Figure 6 are quite good.
- EGRL directly builds upon CERL so it is not very novel, but it has not been applied before to this domain.

Cons:

- The paper evaluates the technique on just 3 workloads. This is in contrast to [1] who evaluate on 372 different workloads and [2] who evaluate on 96 synthetic graphs. [3] and [4] also evaluate on a very small number of workloads, but I believe they probably got a freepass since they were the earliest works in their domain.
- The baseline that the experiments are being compared against might be weak - in figure 3, it looks the policy from both EGRL and EA in iteration 0 itself beat the baseline for Resnet-101 and BERT!
- How long does it take to perform one iteration? And how long does it take to train the policy? This would be useful to get an idea of how EGRL fairs against [3] and [4] which also trained on real hardware and took many hours to finish training the policy.
- The demonstration of generalizability is insufficient - it is difficult to conclude that EGRL can generalize to other workloads. For eg. in Figure 4 (left), the policy performs worse than the baseline for Resnet-50. Moreover, two of the workloads are from the Resnet family.
- If it takes a long time to train each policy and if the model also shows poor zero-shot generalizability, it makes me question if this approach is practical for a compiler setting where a user would typically want the compilation to be completed quickly.

Overall:

I felt that the paper has some interesting ideas but needs more experiments.


Questions and Clarifications:

- I believe that the related work section should add a clarification - [1], [2], [3] and [4] primarily deal with device placement, i.e., placing components of computation graph on different CPUs/GPUs to optimize run time via better parallelization. While this work is concerned with mapping components to different memory hierarchies on the same device.

- While EGRL's action space is larger than [5], the action space in [1] is much larger - for a graph with 2000 nodes to be placed on 2 devices, there are 2^2000 possible choices ~ 10^603

- In Figure 6 (Bottom), is there any reason why you didn't show the result for BERT?


References:

[1] Aditya Paliwal,  Felix Gimeno,  Vinod Nair,  Yujia Li,  Miles Lubin,  Pushmeet Kohli,  and Oriol Vinyals. Reinforced genetic algorithm learning for optimizing computation graphs. arXiv preprint arXiv:1905.02494, 2020

[2] Ravichandra Addanki, Shaileshh Bojja Venkatakrishnan, Shreyan Gupta, Hongzi Mao, and Moham-mad Alizadeh. Placeto: Efficient progressive device placement optimization. In NIPS MachineLearning for Systems Workshop, 2018

[3] Azalia Mirhoseini, Anna Goldie, Hieu Pham, Benoit Steiner, Quoc V. Le, and Jeff Dean. A hierarchical model for device placement. In International Conference on Learning Representations, 2018. URL https://openreview.net/forum?id=Hkc-TeZ0W.

[4] Azalia Mirhoseini, Hieu Pham, Quoc V. Le, Benoit Steiner, Rasmus Larsen, Yuefeng Zhou, NaveenKumar, Mohammad Norouzi, Samy Bengio, and Jeff Dean. Device placement optimization with reinforcement learning. arXiv preprint arXiv:1706.04972, 2017.

[5] Azalia Mirhoseini, Anna Goldie, Mustafa Yazgan, Joe Jiang, Ebrahim Songhori, Shen Wang, Young-Joon Lee, Eric Johnson, Omkar Pathak, Sungmin Bae, Azade Nazi, Jiwoo Pak, Andy Tong, KavyaSrinivasa, William Hang, Emre Tuncer, Anand Babu, Quoc V. Le, James Laudon, Richard Ho,Roger Carpenter, and Jeff Dean. Chip placement with deep reinforcement learning. arXiv preprint arXiv:2004.10746, 2020.

---

> ### Author Response · Authors · 2020-11-13
> **Response 2 of 2**
>
> **"If it takes long to train and has poor generalization - why is it practical?":**
> EGRL is practical because it allows a designer to optimize run-time on hardware with minimal understanding of compiler operations or memory hierarchy.
>
> We respectfully disagree with the notion that it is not practical because “it takes long to train”. Prior works with faster training times noted above do not train on hardware directly. As discussed, some of them do not model true hardware constraint - and thus might need additional fine-tuning on hardware for peak performance. This is unknown as those works do not map their solutions to hardware.
>
> Also, as noted, training time for EGRL scales ~linearly with parallel nodes. This could be particularly useful when optimizing on large clusters.
>
> **Related works:**
> All of these related works are discussed in the paper and we do note that these aim to “optimize the execution of computation graphs on hardware”. Based on your suggestion, we have added additional verbiage noting the differences in the problem domains.
>
> **Action space in REGAL [1] compared to EGRL:**
> Regal trains and tests on a dataset of graphs with varying numbers of nodes. In their Fig 5, they show the distribution of the number of nodes in a dataset. Their publicly released “synthetic dataset” comprises graphs with at most 200 nodes - thus making the action space 2^200 or ~10^60. This is much smaller than our action space which goes up to 10^358. The equivalent graphs for them would have to be ~1200 nodes.
>
> Their TF memory graphs dataset does go up to 2000 nodes. However, the distribution of the number of nodes is centered around 100 with the 1200+ nodes falling on the tail of the distribution. Thus the majority of their graphs are several orders of magnitude smaller than what we consider.
>
> Pertinently, they only report mean performance for their evaluations in Fig 2 and Fig 3 on a test set that samples graphs from this distribution. From this, it is difficult to conclude if their reported mean performances would actually scale to 2000 nodes. They show performance variance in the Appendix - but do not isolate the performance for 1200+ nodes - which would be roughly equivalent to our action space.
>
> Having said that, our future work will investigate scaling EGRL to extremely large graphs with ~1000+ nodes. So far, for 350+ nodes, we haven’t observed any noticeable performance degradation with size.
>
> **Visualization for BERT mappings:**
> We left the BERT mapping out due to constraints on space. We see similar mapping differences in BERT as for the ResNet mappings. We have added this to the appendix in the revised manuscript - and will update it in the main paper in the camera-ready version

---

> > ### Comment · AnonReviewer1 · 2020-11-17
> > **Comments**
> >
> > I thank the authors for responding to my questions.
> >
> > *Contrast to REGAL that evaluates 372 workloads and Placeto that evaluates 96 synthetic graphs*:  I acknowledge that validating results on real hardware is a big win for EGRL over [1] and [2], however, my comment was a critique on the amount of evidence provided that EGRL works on real hardware. As it stands, we are just two data points away from a sample size of one :) I understand the computational limitations of these kind of experiments, however, my judgment tells me that an excellent version of this paper would have evaluated the technique on many more deep learning models (GANs, RNNs, Graph Neural Networks, maybe even random neural networks).  I stick to my stance that the performance of EGRL looks promising but should be taken with a grain of salt until more evidence is provided.
> >
> > *If it takes long to train and has poor generalization - why is it practical?*: Thanks for the clarification - if I understood correctly, the practicality is similar to that of [3] and [4].
> >
> > *Action space in REGAL [1] compared to EGRL* - Good point, while [1] does have some graphs with larger action spaces, they only provide the average performance of REGAL and it is hard to conclude if that translates to good performance on large graphs.
> >
> > *Baselines might be weak, Training time, Generalization properties, Related works, Visualization for BERT mappings* - Thank you for the clarification
> >
> > Overall, I am happy to bump my score to a 6.
> >
> > [1] Aditya Paliwal, Felix Gimeno, Vinod Nair, Yujia Li, Miles Lubin, Pushmeet Kohli, and Oriol Vinyals. Reinforced genetic algorithm learning for optimizing computation graphs. arXiv preprint arXiv:1905.02494, 2020
> >
> > [2] Ravichandra Addanki, Shaileshh Bojja Venkatakrishnan, Shreyan Gupta, Hongzi Mao, and Moham-mad Alizadeh. Placeto: Efficient progressive device placement optimization. In NIPS MachineLearning for Systems Workshop, 2018
> >
> > [3] Azalia Mirhoseini, Anna Goldie, Hieu Pham, Benoit Steiner, Quoc V. Le, and Jeff Dean. A hierarchical model for device placement. In International Conference on Learning Representations, 2018. URL https://openreview.net/forum?id=Hkc-TeZ0W.
> >
> > [4] Azalia Mirhoseini, Hieu Pham, Quoc V. Le, Benoit Steiner, Rasmus Larsen, Yuefeng Zhou, NaveenKumar, Mohammad Norouzi, Samy Bengio, and Jeff Dean. Device placement optimization with reinforcement learning. arXiv preprint arXiv:1706.04972, 2017.

---

> > > ### Author Response · Authors · 2020-11-19
> > > **Thanks | Agreed about more data points**
> > >
> > > We agree more data points would make EGRL's claims even more concrete. Since we were limited by hardware constraints, we had to take a call on when we had sufficient data for a proof of concept and felt that the three representative large models were a reasonable data point.
> > >
> > > We appreciate your revisiting the score!

---

> ### Author Response · Authors · 2020-11-13
> **Response 1 of 2**
>
> Thank you for your detailed and insightful review. We would like to address your main concerns below.
>
> **Contrast to REGAL that evaluates 372 workloads and Placeto that evaluates 96 synthetic graphs:**
> We contrast our work with REGAL [1] in the paper - which assumes zero latency and infinite bandwidth that is not practical on actual hardware. Specifically, they mention “We consider two tasks, one is minimizing peak memory and the other is minimizing running time, both on two homogeneous devices with 16 GiB of memory each and synchronous tensor transfers with zero cost (zero latency and infinite bandwidth).”
>
> We also address comparisons with Placeto [2] in our related works section and note the significant amount of heuristic, domain specific grouping utilized as pre-processing steps. In contrast EGRL assumes no domain knowledge and directly handles the entire combinatorial action space. Placeto, too, trains on a simulator and not on physical hardware. Specifically, they mention “Since it can take a long time to execute placements on real hardware and measure the elapsed time, we built a reliable simulator that can quickly predict the runtime of any given placement for a given device configuration.”
>
> The key differentiating point of EGRL is that our pipeline is entirely trained and validated on a hardware accelerator end-to-end. With such real-world constraints, training time for each policy is gated by real bandwidths and memory constraints. In fact, the key point  of EGRL is to learn the trade-off between the relative memory capacity and bandwidths of the memory components on the chip. Thus our solution requires no further sim2real steps to transfer to actual hardware - whereas both of these approaches would need to be validated on hardware. Neither of these papers address if their solution does indeed transfer to real hardware.
>
> **Baselines might be weak:**
> EGRL uses a population based approach with ranking. So even at the zero-th iteration, there are a large number of candidate solutions of which we report the champion performance. Further, the mapping logic in the compiler comprises standard heuristics like cache eviction strategies that are common across processor architectures. We also compare against more well understood baselines like dynamic programming and pure policy gradient RL.
>
> Our choice of baselines was primarily motivated by prior work in this area. For example, Mirhosseini 2017, 2018, 2020, Placeto, REGAL, etc do not provide official source code. Further, each of these prior works have domain-specific, heuristic sub-modules that are not guaranteed to work in a different domain. However, all of these papers have a core RL component - and thus we chose to benchmark using SAC - which is a state of the art RL algorithm - as one of the baselines. We contend that this choice distils the core RL aspect across all of these prior works.
>
> **Training time:**
> Our policies train in order of hours if restricted to a single hardware node. The main cost of each iteration is the rollout step - which is essentially a forward inference pass using the workload. The update cost for the policy is negligible in comparison. When trained on multiple hardware nodes, we can take advantage of the parallelization inherent in EGRL and training time reduces roughly linearly with an increasing number of nodes.
>
> **Generalization properties:**
> We completely agree that the generalization experiments cannot be used to conclude that an EGRL policy can fully transfer from one workload to another. In fact, we concede this in the paper. The scope of our generalization experiments is to investigate if the learnt GNN embedding are performant at all when transferred zero-shot. We found that zero shot transferred policies can be somewhat performant - although they are definitely sub-optimal compared to training from scratch.
>
> In Fig 4, the policy trained on BERT transferred to ResNet-101 and achieved a ~25% speedup. As you note, it did not achieve any speed-up when transferred to ResNet-50. The policy trained on ResNet-50 transferred to ResNet-101 and achieved ~50% speedup. It also transferred to BERT and achieved ~25% speedup. As we acknowledge in the paper, further fine-tuning is required to bridge the performance gap - however, the fine-tuning should consume fewer samples than training from scratch. We defer a full study of the generalizability properties to future work.

---

### Official Review · AnonReviewer2 · 2020-10-27
**Official Blind Review #2**

**Rating:** 7
**Confidence:** 4

**Review:**

The paper proposes Evolutionary Graph Reinforcement Learning to solve the memory placement problem. Main ideas are using GNN as the network architecture for reinforcement learning agents that look for more informed priors for evolutionary algorithms. Overall novelty of the paper comes from the neat combination of RL, EA, and GNN, and applying it to memory placement (ML for Systems).

The paper indeed tackles an important problem that can affect the overall performance and efficiency of the hardware. I believe the reorganization of various off-the-shelf ML techniques to solve real problems in the systems domain marks a large contribution, hence the positive overall rating.

One of the main drawbacks of the paper is that the paper only tests on a single type/configuration of hardware. While this is fine to some extent, this makes it hard to get confirmation about the generality of the overall method considering the large variance of the speedup.

Another related question comes from how this work relates to the optimizations of the dataflows [1,2]. As it is difficult to evaluate the overall memory communication without considering the order of operations, etc. the work in turn neglects the big question and focuses on only the partial view of the problem. It would provide a nice reference point if some of these points are discussed in the paper.

Last question comes from the baselines. While the previous works on tensor optimizations [3,4] are very closely related and many of the ideas provide a good comparison point, these have not been discussed nor cited. For example, I guess AutoTVM's way of approximating the search space using TreeGRU or XGBoost can help. Also, Chameleon's way of sampling the examples using adaptive sampling may provide an interesting reference point in terms of reduction of number of samples.

Overall, I have enjoyed reading the paper and I find the ideas in the paper interesting. I am currently weakly pro for the paper, and look forward to the authors' response :)

Questions
1. Could you provide and overview of the NNP-I's architecture in the appendix? Also, possibly ablation studies over different configurations of the hardware.
2. What are the relative communication speed of SRAM, LLC, and DRAM?
3. It is well discussed in the computer architecture community that the memory communication is very much determined by the dataflows of the architecture. How are the results affected by these dataflows?
4. How does the work compare to the methods described in [3,4]?

[1] "Eyeriss: A spatial architecture for energy-efficient dataflow for convolutional neural networks", ISCA 2016

[2] "Interstellar: Using Halide's Scheduling Language to Analyze DNN Accelerators", ASPLOS 2020

[3] "Learning to optimize tensor programs", NeurIPS 2018

[4] "Chameleon: Adaptive code optimization for expedited deep neural network compilation", ICLR 2020

---

> ### Author Response · Authors · 2020-11-13
> **Response 1 of 1**
>
> Thank you for your review. We would like to address your main concerns below.
>
> NNP-I architecture and configurations:
> As suggested, we will add a section in the Appendix detailing the NNP-I architecture. While the NNP-I does allow for different configurations to change power consumption via frequency control, the results reported in the paper are for one configuration.The results in our paper, however, come from running experiments on different instances of the hardware. For those cases, while the absolute measured throughput varied, as expected, from chip-to-chip, the observed relative improvements over the baselines had little variability.
>
> We defer a formal study across several hardware configurations to future work.
>
> Relative communication speeds of SRAM, LLC and DRAM:
> The memory size and bandwidth trade-offs between SRAM, LLC and DRAM are provided in section 3; LLC (24MB) and SRAM (4MB) are ~10x and 100x faster than DRAM (32GB) respectively.
>
> Effect of dataflows on memory communication:
> We address this point while justifying our choice of a graph representation for the incoming workload. The edges in the graph input to the GNN agent captures the relative sequences of operations as each node in the input graph represents an operational layer. The advantage of the GNN representation is that this raw graph representation is then mapped to lower dimensional GNN embeddings - specifically the hidden layers of the Graph UNet. These GNN features, in theory, should be invariant representations of the data-flow - including the order of operations.
>
> In contrast a sequential mapping strategy necessarily has to prioritize some nodes over others (typically by mapping the first node and working towards the end of the incoming workload, though this is absolutely not necessary). In practice though we must map all nodes in order to even achieve a valid mapping and there is no notion of order in the final complete mapping. EGRL simply maps all nodes simultaneously (in one step).
>
> We do not quantify this directly - however, we note that such invariant feature representations are commonly noted in hidden layers of deep neural networks in general. Thus, we feel that the GNN approach actually does capture the full view of the problem - as opposed to sequential mapping strategies which have a partial view of the problem for a given iteration. While our overall speedup metric provides an aggregate view of memory communication for the entire workload, a rigorous study investigating this at the node level is an interesting angle for future work.
>
> Comparison to [3] and [4]:
> We will add these references to the manuscript.
> In AutoTVM [3] and Chameleon [4], the authors investigate the automatic optimization of tensor operations for arbitrary hardware targets. AutoTVM builds a statistical model to estimate the cost of each low-level program. The specific implementations deploy gradient-boosted trees and TreeGRU - the latter being a deep-learning based method. Chameleon takes an RL based approach. Although these cannot be directly mapped to a “device mapping” strategy, they can be considered similar in nature to the problem of optimizing in a large combinatorial space - similar to our paper - and either method could potentially be applied to the device mapping problem with appropriate adjustments for the state space.
>
> Chameleon is obviously closer to our work - and uses a purely actor-critic network. However, here too, the “leverage a clustering algorithm to find configurations that are representative of each cluster”. This is consistent with other pure RL based prior-work that rely on clustering algorithms. We could not glean from the paper the magnitude of the combinatorics problems they solve - however, they report their results on AlexNet, VGG-16 and Resnet-18 which are all significantly smaller workloads compared to the ones in our work or even other related works.

---

> > ### Comment · AnonReviewer2 · 2020-11-15
> > **Questions**
> >
> > I understand that a formal study across several hardware configurations is not feasible during the rebuttal period. However, I would like to see some reference to the details of the architecture for more informed evaluation of the paper.
> >
> > Regarding the answer about the dataflows, I think there may be a little mismatch between the definition of dataflows. The dataflows that I refered to is not limited to just the sequence of the operations. The dataflows I am referring to includes the mapping of the operations to the hardware which one incarnation would be tiling. I believe NNP-I would not have a computing array that computes the entire layer in a single step, but relies on some tiling which is done by the compiler. Without encoding these details to the GNNs, it seems to me that the current work only takes a partial view.
> >
> > This dataflow issue also has to do with [3,4]. In my original review, I was looking for authors' view in relation to how the results presented in the paper would be affected by different dataflows, sequencing, etc. [3,4] shows that there is a significant variance in the inference speed of deep networks. Therefore, the overall gains from the proposed method can vary significantly depending on how optimized the paper's baseline is. I would like to see some analysis and evaluation with regard to this.
> >
> > From the authors' response, it seems that the authors are claiming that this method encodes the relative sequence through the GNN. Does this mean that the work would provide similar speedup regardless of the sequence of operations? For example, in [5] it is shown that the overall memory footprint is significantly affected by the sequence of the operations, and the paper presents an optimal sequence of operations. My question is whether this method would provide the same significant gains shown in the paper even after the "wasteful" memory communication from the suboptimal sequence of operations have been obviated by [5].
> >
> > [5] "Ordering chaos: Memory-aware scheduling of irregularly wired neural networks for edge devices", MLSys 2020

---

> > > ### Author Response · Authors · 2020-11-17
> > > **NNP-I architecture | EGRL's partial view of the problem | Discussion on graph re-writing [5]**
> > >
> > > Thank you for your thoughtful response and for taking the time to clarify the definition of data flows.
> > > The updated manuscript now has an additional section on the architecture of NNP-I. Deeper details can also be found in [https://en.wikichip.org/wiki/intel/microarchitectures/spring_hill].
> > >
> > > Broadly, NNP-I consists of 12 ICE (inference compute engine) cores each with a fast 4MB SRAM. All ICEs have access to a shared, slower LLC (24MB). Additionally, an even slower 32GB DRAM can also be accessed by an ICE core via the LLC. Thus, our agent needs to learn to trade off between memory capacity and bandwidth across these options.
> > >
> > > In EGRL, the GNN **only** encodes the input workload explicitly. In this work, our RL agent, and other baselines, only specify if a layer should be placed in one of the three memory types. The choice of the specific ICE core (and therefore the specific Deep SRAM address) is left to further cache management logic in the compiler. Since EGRL trains itself on the resultant latency, one could argue that it learns to model the consequence of the additional compiler optimizations implicitly - but this is never modeled explicitly.
> > >
> > > Thus, EGRL **does have a partial view of the problem** in the context of data flows as you describe it.
> > >
> > > Having said that, it would be possible to apply EGRL on an action space larger than what we chose: E.g. To let it control the number of compute kernels (ICE) to use per thread or per op, scheduling between branches, and more. We believe that the greater the complexity of the chosen action space is, the greater the potential of optimization but also the convergence time of the algorithm and need for specific knowledge of the underlying architecture. These options are recorded for future work.
> > >
> > > We cannot say conclusively whether EGRL would provide similar speedup regardless of the sequence of operations executed by the compiler. A more rigorous way to answer that question is to test EGRL on a wide range of hardware/compilers for a given computational graph. That is certainly something worth investigating in future work.
> > >
> > > The question about the optimality of the baseline is fair. The work in [5] specifically aims to reduce wasteful memory communication resulting from “irregularly wired neural networks” that stem from “random network generators” and “neural architecture search”. In our work, we considered three very commonly adopted networks (Resnet-50, Resnet-101 and BERT) - all of which could be generally considered optimally wired compared to a randomly discovered topology. Thus, we do not think that an approach like [5] would find significant memory bandwidth savings compared to standard compiler optimizations like layer fusion.
> > >
> > > You raise another interesting question - if EGRL would provide the same gains on top of the optimizations stemming from [5]. The work in [5] performs hardware independent optimization of the network topology in a way that minimizes “wasteful” sequences -- essentially a form of graph rewriting. On the other hand, EGRL aims to optimize on a hardware specific performance metric while executing a computational graph on the target - but has no ability to modify the topology of a computational graph. Thus, the two are complementary approaches to accelerate the execution of such graphs.
> > >
> > > Consider the scenario where a computational graph optimized by [5] is to be executed on a target hardware. Depending on the application, the desired hardware performance metric could be very different - e.g., pure throughput (as we use in the paper) or throughput per watt. Additionally, depending on the hardware configuration, various compute and memory components may have very different trade-offs. Hardware specific constraints like these are not handled by [5] at all. EGRL will (in theory) learn to optimize the application specific performance metric under the hardware-specific constraints. An interesting future work could be hardware aware graph-writing - which could investigate if the graph re-writing could be jointly optimized with memory mapping for a given hardware target for neural architecture search.

---

> > > > ### Comment · AnonReviewer2 · 2020-11-17
> > > > **Comments**
> > > >
> > > > Thank you for the detailed and thoughtful response.
> > > > In general, I am satisfied with the authors' response, and I would be happy to vote strongly for acceptance **on the premise that the authors revise the manuscript so that it includes description of the work in the context of the dataflows and the other ML-based code optimizations.**  I believe this work can be fully appreciated only when connections to other aforementioned aspects of compilation and optimization are provided together.
> > > >
> > > > The work provides interesting insights and a good practical application of GNNs and its marriage with evolutionary algorithms. While there are some weaknesses in evaluation as the Reviewer 1 and 3 pointed out. However, I believe the insights and the gains provided in this paper outweighs such weaknesses.

---

> > > > > ### Author Response · Authors · 2020-11-17
> > > > > **Manuscript updated with discussion on data flows and relation to other ML-based code optimizations**
> > > > >
> > > > > We thank you again for the engaging discussion.
> > > > >
> > > > > Based on your feedback, the updated manuscript now contains an extended discussion in Appendix G on the relevant architectural details of NNP-I and details on the memory hierarchy and trade-offs. We also added to it a subsection titled `EGRL's Interaction with NNP-I` where we go into detail about the exact level of control EGRL has on the data flow in the hardware.
> > > > >
> > > > > In order to retain the original section numbering and 8-page limit while it is under review, we added this complete section to the Appendix. Since the final manuscript allows for 9 pages, we will move this section to the main paper in the camera-ready version.
> > > > >
> > > > > Specifically, we clarify that while EGRL can be extended to control various hierarchy of the data flow, we choose to study the least restrictive formulation (i.e., requiring the least amount of domain knowledge) where downstream cache management and other optimizations are left to the compiler.
> > > > >
> > > > > We added an additional paragraph on the complementarity of EGRL with other ML based graph optimization methods such as [5] but also other forms such as quantization and model compression which are generally not hardware-aware.
> > > > >
> > > > > We hope these changes address the points you raised in this discussion.

---

> > > > > > ### Comment · AnonReviewer2 · 2020-11-19
> > > > > > **Comments**
> > > > > >
> > > > > > Overall, I feel the revision seems fine to me. As a last comment, I would like to see the reference to the AutoTVM, Chameleon that addresses the code optimization side of the work as I mentioned in the original review. Overall, I am satisfied with the authors' response, hence increased my score from 6 to 7.

---

### Official Review · AnonReviewer4 · 2020-10-28
**Interesting extension of the Graph Optimization using DRL line of work**

**Rating:** 7
**Confidence:** 4

**Review:**

Optimizing the execution of deep neural networks has tremendous impact on the cost and performance in many industries due to the proliferation of "Deep Learning". There has recently been an interesting line of work of using learning to optimize policies related to placement and scheduling of the neural network computation graph outperforming tediously hand-crafted heuristics. The proposed paper would be a nice extension along this line.
The impact of memory placement for DNN has been clearly motivated and is easy to appreciate.
The paper overall is clear and easy to follow. The methodology is sound and justified. Experiments and results make a compelling case in supporting the claims.

I think one aspect that is insufficiently motivated is the need to use a hybrid approach between RL and evolutionary algorithms. Results show improvement in performance but it is not clear to me why. Perhaps this is addressed in the CERL paper which I am not familiar with.

In the results section, it would be nice to see the sample complexity of each of the methods. I see that the number of iterations are shown in Figure 3 but it is not clear to me if that also corresponds to the number of samples consumed by each of the approaches before they converge.

---

> ### Author Response · Authors · 2020-11-13
> **Response 1 of 1**
>
> Thank you for your review. We would like to address your main concerns below.
>
> **Need for the use of a hybrid approach:**
> The use of RL and EA is motivated by several works in recent literature that have demonstrated the effectiveness of combining reinforcement learning with search for extremely large combinatorial problems.
>
> For example, AlphaGo and AlphaZero successfully combined RL and look-ahead search (MCTS) to find effective strategies on board games. CERL showed similar gains in large action spaces for continuous control by combining RL and EA search. We adopted the EA search due to ease of implementation - but the broader design could also have utilized tree search as well as other search methods.
>
> Sparse rewards are a common problem in RL and our  case is no different: A sparse reward obtained once for all nodes renders it difficult for RL to learn in isolation. This is alleviated by EA, as it relies only on episodic performance. On the other hand, EA converges extremely slowly compared to RL. EGRL provides a framework for the slower EA component to anchor its search around partial solutions provided by the RL component. This speeds up EA while retaining its ability to find stable and high performance solutions. This also avoids the problem of reward shaping, which is commonly used in sparse reward problems. We integrate the sparse episodic feedback natively into the reward.
>
> Further, the population based approach in EA allows for significant asynchronous parallelizability since all policies can roll-out in the environment completely independently of each other. Since the rollouts dominate the compute budget, convergence time for EGRL decreases with a larger number of hardware instances available.
>
> The importance of the RL component to improve on top of EA is shown in the ablation studies in Fig 3 where we see that the combined EGRL formulation consistently outperforms the RL and EA components in isolation. This quantifies our assertion that both EA and PG are essential components of EGRL.
>
> **Sample complexity:**
> The number of iterations reported is the same as the sample complexity in this paper. We have updated the text in the revised manuscript to make this clearer.

---

### Official Review · AnonReviewer3 · 2020-10-29
**Well-written paper on memory mapping method that outperforms native NNP-I compiler by 28-78%**

**Rating:** 5
**Confidence:** 5

**Review:**

The paper describes a machine learning method for mapping computational nodes in a neural network onto different levels of the memory hierarchy (DRAM, LLC, and SRAM) to minimize latency of inference. The proposed approach builds on CERL, combining policy gradient, reinforcement learning, and graph neural network, achieving 28-78% speed-up over the native NNP-I compiler on vision and language benchmarks (ResNet-50, ResNet-101, and BERT).

Overall, the paper was well-written, targets an impactful problem, and the reported improvements (28-78% over native compiler) are impressive.

In the related work section, I did have a concern, as the authors state “For example, previous work with manual grouping (sic) operate at most in 5^280 \~= 10^196 dimensional action space (Mirhoseini et al., 2020), compared to 10~ 20^358 for our BERT problem”. However, Mirhoseini et al., 2020 (“Chip placement with deep reinforcement learning”) places “a few thousand clusters” (>=2000 nodes) onto a grid with “an average of 30 rows and columns” (~900 cells), so wouldn’t the action space be at least 900^2000? Also, didn’t that work use a heuristic grouper (hMETIS), but maybe that’s close enough to “manual”?

The authors only look at three benchmarks, but they were well-chosen (two representative vision models and one large language model). It’s also good that they compare against PG and EA alone as a form of ablation, given that their method is effectively a combination of these two. It would have been better if they also had compared with prior state-of-the-art (e.g. HDP, REGAL, Placeto, or (the unmentioned) GDP / GO), but it is somewhat understandable given that their code does not seem to be open-sourced.

I liked that the authors report mean and standard deviation for the five runs, and measured “true” reward by running on hardware. I also thought they did a good job motivating their method (aside from the questionable statements about action spaces in prior work), and of analyzing and visualizing its performance.

Nits:
In the Method section, “the compiler rectifies them and outputs a modified map, M_c, that is fully executable (Line 6).” It would probably be good to add “in Algorithm 1” so as not to confused the reader.

“It comprises of a single PG learner” -> “It is comprised of…”

“Both methods are known to produce highly performant and stable solutions but are also significantly slow compared to Deep RL” (“significantly slower than”?)

“While the transferred policies are clearly underperform those from scratch” -> “underperforming”

---

> ### Author Response · Authors · 2020-11-13
> **Response 1 of 1**
>
> Thank you for your review. We would like to address your main concerns below.
>
> **Sizes of the action space in prior work:**
> The reference in our paper should have been to Mirhosseini et al 2017 (Device Placement) and not Mirhosseini et al 2020 (Chip Placement). We have corrected this error in the updated manuscript.
>
> As discussed in our paper, a number of prior works deploy heuristic pre-processing steps to reduce the action space for the learning problem. Since it is difficult to compare heuristics across different applications, we focus primarily on the complexity of the learning problem. We feel this is a fair approach since EGRL does not rely on any algorithmic heuristics and requires only basic domain knowledge e.g., the state and action space dimensions to initialize the policy networks and the task objective to optimize.
>
> In the 2017 paper (our intended reference), Table 1 shows the workloads considered. The largest number of heuristically derived groups is 280 corresponding to NMT. The maximum number of devices considered for each group of operations was 5 → thus leading to the maximum combinatorics space of 5^280 or 〜 10^196 as discussed in our paper.
>
> In the 2020 paper (the incorrect reference), as you note, the overall combinatorics is extremely large. However, the problem for the reinforcement learning agent in that work is significantly smaller than that. For example, their paper notes that the action space is all valid placement of one macro. Following the RL-based macro placements, the standard cells are placed using classical force-directed methods rather than any learnt policy.
>
> Thus if we focus on the RL problem, the action space is all possible placement choices for one macro - which in their case are the centers of “a few thousand grid cells”. Since they do not take a joint action across all macros - placing macros one at a time (Figure 1a) - their actual action space is of the order of “a few thousand”. They also do not report the total number of macros placed - thus making it difficult to ascertain the overall scale of the combinatorics problem.
>
> In Section 2, we provide some trade-offs between sequential placements and simultaneous placements. We will update it to discuss the action space comparison with this paper in the final manuscript.
>
> **Comparisons with HDP, Regal, Placeto etc.:**
> We appreciate that you recognize that direct comparisons with HDP, REGAL, Placeto etc. are difficult due to the lack of official open-source code. Additionally, each of these works rely on domain specific heuristic pre-processing steps - which makes it difficult to directly compare on different problem domains. We therefore chose to compare against the state-of-the-art RL baseline (SAC), since policy-gradient based RL is generally a common choice for the learning portion of these works.
>
> **Nits**: Thank you for pointing these out. We have fixed these typos in the updated manuscript.
>
> We hope the above clears up the confusion around action spaces in prior work. Since you score our paper marginally below the acceptance threshold, we would greatly appreciate your feedback if you feel other aspects of the paper require more detail.

---

### Decision · Program_Chairs · 2021-01-07
**Final Decision**

**Decision:**

Accept (Poster)

**Comment:**

Most of the reviewers agree that this paper presents interesting ideas for an important problem. The paper could be further improved by having a thorough discussion of related works (e.g. Placeto) and construct proxy baselines that reflect these approaches.


The meta-reviewer decided to accept the paper given the positive aspects, and encourages the author to further improve the paper per review comments.


Thank you for submitting the paper to ICLR.